# Advanced Zinc–Magnesium Alloys Prepared by Mechanical Alloying and Spark Plasma Sintering

**DOI:** 10.3390/ma15155272

**Published:** 2022-07-30

**Authors:** David Nečas, Ivo Marek, Jan Pinc, Dalibor Vojtěch, Jiří Kubásek

**Affiliations:** 1Department of Metals and Corrosion Engineering, Faculty of Chemical Technology, University of Chemistry and Technology Prague, Technická 5, 166 28 Prague, Czech Republic; mareki@vscht.cz (I.M.); vojtechd@vscht.cz (D.V.); 2Department of Functional Materials, Institute of Physics of the Czech Academy of Sciences, Na Slovance 1999/2, 182 21 Prague, Czech Republic; pincik789@gmail.com

**Keywords:** zinc, magnesium, mechanical alloying, powder metallurgy, mechanical properties

## Abstract

Zinc and its alloys are considered as promising materials for the preparation of biodegradable medical devices (stents and bone fixation screws) due to their enhanced biocompatibility. These materials must achieve an ideal combination of mechanical and corrosion properties that can be influenced by alloying or thermomechanical processes. This paper presents the effects of different mechanical alloying (MA) parameters on the composition of Zn-1Mg powder. At the same time, this study describes the influence of preparation by MA on Zn-6Mg and Zn-16Mg alloys. The selected powders were compacted by the spark plasma sintering (SPS) method. Subsequently, their microstructures were studied and their mechanical properties were tested. The overall process led to a significant grain refinement (629 ± 274 nm for Zn-1Mg) and the formation of new intermetallic phases (Mg_2_Zn_11_, MgZn_2_). The compressive properties of the sintered samples were mainly related to the concentration of the alloying elements, where an increase in concentration led to an improvement in strength but a deterioration in ductility. According to the obtained results, the best properties were obtained for the Zn-1Mg alloy.

## 1. Introduction

Biomaterials are artificial or natural materials used for the manufacturing of structures, which replace damaged or lost human tissue in order to restore its previous function [1]. These medical devices can be divided into three groups according to their interaction with the human body. Bioinert materials (Ti-6Al-4V, Co-28Cr-6Mo, 316 L stainless steel) are characterized by excellent corrosion resistance and a relatively high modulus of elasticity. However, such materials interact insufficiently with tissue and the high modulus also promotes a stress shielding effect. This can ultimately lead to implant loosening. Bioactive materials (surface-modified titanium alloys, bioceramics, hydroxyapatite) interact with the surrounding tissue, which improves regeneration and bonding with the implant. Finally, biodegradable materials (Fe-, Mg- and Zn-based alloys) gradually dissolve in the body and completely degrade once their function has been fulfilled [2,3,4,5,6]. Among these groups, the last group seems to be the most intensively studied with an interest in finding suitable materials with appropriate properties. Great interest lies particularly in zinc and its alloys.

Zinc is an essential element for various functions in the human organism and has a reasonable corrosion rate. Compared to Mg and Fe, its corrosion processes are not associated with the formation of hydrogen or toxic corrosion products, respectively. Therefore, zinc and its alloys have been studied in the last decade as promising biodegradable materials [7].

However, the mechanical properties of pure zinc are insufficient for such applications. The tensile strength of zinc in the as-cast conditions is approximately 20 MPa and the elongation to failure reached only 0.2% [8]. Therefore, various alloying elements (Mg, Mn, Ca, Sr) are generally considered to improve the mechanical properties [9]. Additionally, materials are often thermomechanically processed by extrusion, rolling or drawing [7,10]. The properties of wrought zinc alloys are significantly different from as-casted materials due to the lower grain size, specific textures and redistribution of intermetallic phases in the microstructure [8].

Magnesium is considered as the most popular alloying element of zinc due to its excellent biocompatibility and good strengthening effect related to the formation of intermetallic phases with zinc [11]. The maximum solubility of Mg in Zn is approximately 1 wt.% at eutectic temperature, but decreases to almost zero at ambient temperature [12]. The addition of magnesium leads to the formation of intermetallic phases such as Mg_2_Zn_11_ (6.5 wt.% of Mg) or MgZn_2_ (16 wt.% of Mg) [13]. These precipitates promote higher strength of materials but at the same time reduce their ductility. As a consequence, Zn-1Mg is considered as the “gold standard” of biodegradable zinc-based alloys [14,15,16]. The material in the as-cast state consists of relatively coarse zinc dendrites surrounded by a eutectic mixture, but after hot rolling or hot extrusion the average grain size is reduced due to the recrystallization to values of about 2–5 μm [17]. In addition, the brittle intermetallic phases generally break and redistribute in rows parallel to the direction of deformation. However, the improvement of the materials’ properties by the proposed methods is limited. The grain size cannot be further decreased due to the low recrystallization temperature of pure zinc and also the size and distribution of the intermetallic phases is more or less given [17]. Finally, these materials generally develop strong textures that increase the anisotropy of the mechanical properties [14,17,18,19].

Powder metallurgy (PM) techniques offer some possibilities to affect the mechanical properties by further decreasing both the grain size and the size and distribution of intermetallic particles [5,20,21]. Another important factor is the possibility to increase the solid solution solubility of the alloying elements in the matrix above equilibrium conditions by the application of mechanical alloying. This promotes the formation of metastable microstructures with a direct effect on mechanical and corrosion properties [22,23,24]. However, powder metallurgy techniques including mechanical alloying are less commonly used for zinc-based alloys [25,26,27,28] compared to other metals. Precursors in the form of powders can be compacted by various methods, including extrusion [29], hot isostatic pressing [30], hot vacuum pressing [31], etc. In this work, spark plasma sintering (SPS) [32,33] was chosen because rapid compaction methods bring the possibility of preserving the fine-grained structure of the precursor powder.

It is worth mentioning that additive manufacturing (AM) methods became very popular in the last years. The reason for that is covered in the ability of the methods to prepare products with the desired shape without further processing. The low melting (420 °C) and boiling (907 °C) temperature of pure zinc [34] causes evaporation during material preparation. This feature is a limiting factor for many AM processes. Although pure zinc has been successfully prepared by selective laser melting/laser powder bed fusion (SLM/LPBF) [34,35] and fused deposition modeling (FDM) [36], gas evaporation during preparation resulted in the formation of pores inside the materials structure. This porosity affects the mechanical and corrosion properties of the products [34,35,36].

In the presented paper, we introduce Zn-Mg binary alloys prepared by a combination of mechanical alloying and spark plasma sintering (SPS). The combination of these two powerful techniques enabled us to prepare materials with a nano-sized microstructure. According to the best of our knowledge, this has not been obtained for zinc-based alloys by any other conventional technique.

## 2. Materials and Methods

### 2.1. Materials Processing

Pure zinc powder (99.9%, particle size < 149 µm, Alfa Aesar) and pure magnesium (99.8%, particle size < 44 µm, Alfa Aesar) (Figure 1) were used for the preparation of Zn-1Mg, Zn-6Mg and Zn-16Mg (wt.%) by mechanical alloying. The concentrations of alloying elements were selected intentionally, based on the binary Zn-Mg phase diagram. Zn-1Mg represents a widely used and recommended binary alloy due to its good compromise between strength and ductility [14,15,16]. The composition of Zn-6Mg and Zn-16Mg correspond to the intermetallic phases Mg_2_Zn_11_ and MgZn_2_. Before milling, stearic acid was added into the mixture to prevent the agglomeration of powder particles during milling. The mixed powders were mechanically alloyed under various conditions specified in Table 1. The weight of the powder mixture was 30 g. Mechanical alloying was performed in ZrO_2_ vessels (125 mL) under Ar protective atmosphere (99.95%) in a Retsch E-max milling machine equipped with a water cooling system. The process temperature was kept between 30 and 50 °C and the direction of rotation was changed every 10 min.

During the mechanical alloying, the RPM parameter affects the energy that is generated when two grinding balls collide. This energy can be calculated for E-max mills according to their parameters from the equations for the deformation energy for the collision of grinding balls (*E_C_*)—(1) and for angular speed (*ω_D_*)—(2) [37].
(1)$$E_{C}=\frac{\left[7.66.10^{-2}\cdot R_{d}^{1.2}\cdot \rho^{0.6}\cdot E^{0.4}\right]\cdot d_{b}\cdot \omega_{D}^{1.2}}{\rho_{b}}$$
(2)$$\omega_{D}=\frac{2\cdot \pi \cdot RPM}{60}$$

In these equations, *R_d_* specifies the distance between the middle of the disk to the second one (0.11 m), $\rho_{b}$ is the density of the material of the milling balls (5730 g/cm^3^), *E* corresponds to the modulus of elasticity of the material of the milling balls (1.75·10^11^ Pa), *d_b_* is the diameter of the milling balls (0.01 m), ρ reflects the surface density of the powder that was stuck to the milling balls (approximately 0.1 kg/m^2^) and *RPM* corresponds to the rotation of the milling vessel [37]. The design of the high energy Retch E-max mills is shown in the diagram (Figure 2) and the calculated results are presented in Table 2.

The selected powder mixtures, which had the smallest particle size distribution and the most suitable phase composition (with small particle size and all magnesium converted to intermetallic phases) were compacted by the spark plasma sintering (SPS)–FCT System HP-D 10 under argon atmosphere (99.96%) with graphite tools at 300 °C. After reaching the selected temperature, a pressure of 19.1 MPa was applied for 10 min. The heating rate was 100 °C/min. The compacted samples were prepared from 22.25 g of powder, generating products in the shape of cylinders that were 10 mm in height and 20 mm in diameter. The diagram showing the time profiles of the process conditions during sintering is presented in Figure 3.

### 2.2. Microstructure

The microstructure of materials was studied using an optical microscope (OM-Olympus PME3) and a scanning electron microscope (SEM—TESCAN LYRA) equipped with an EDS analyzer (OXFORD Instruments AZtec). Prior to the observations, the samples were ground using SiC grinding papers (P400–P2000), then polished with diamond paste D2 (UR-diamant, 2 µm) and finally polished on suspension Etosil E (Metalco, 0.06 µm). The average grain size was calculated using image analyses (ImageJ) on several images. For each grain, the Feret′s diameter was measured. The phase composition of the materials was determined by X-ray diffraction. Measurements were made at ambient temperature using the X’Pert3 Powder instrument with Bragg–Brentano geometry equipped by a Cu anode (λ = 1.5418, U = 40 kV, I = 30 mA). The evaluation of the results was performed using HighScore Plus 4.0 and Topas 3 software. The names and reference codes of the evaluated phases in PDF 4 database were Zn (01-078-9363), Mg_2_Zn_11_ (04-007-1412), MgZn_2_(04-008-6026) and Mg (04-013-4129). The details of the microstructure were studied by transmission electron microscope (TEM-EFTEM Jeol 2200 FS, accelerating voltage 300 kV, LaB_6_). The samples for TEM were prepared using the Gatan PIPS polishing system with Ar ions (Gatan, Pleasanton, CA, USA).

### 2.3. Mechanical Properties

The mechanical properties of the prepared materials were investigated by hardness, compression and bending tests. The microhardness HV1 was analyzed on a Future-Tech FM-100 machine with a load of 1 kgf. A minimum of 10 indentations were made, from which the average value was calculated. The compression and 3-point bending tests were performed on 3 specimens with size 3 × 3 × 3 mm^3^ and 3 × 3 × 15 mm^3^. Statistics data were calculated from the results. Both tests were performed on an Instron 5882 instrument at a ram speed of 1 mm/min at ambient temperature.

## 3. Results

### 3.1. Mechanical Alloying

Firstly, the effect of mechanical alloying parameters on the properties of Zn-Mg mixture was studied. The results clearly showed a significant effect of processing conditions on the phase composition and on the shape and particle size of the powder. The morphology of the prepared powders is shown in Figure 4. The powder particles possessed sharp edges and their size varied depending on the specific conditions. The lower RPM and shorter milling time resulted in particles with an average size about 10 µm. With increasing RPM, the deformation energy generated in the mills increased, promoting cold welding, and producing larger particles with an average size close to 60 µm. Also, significant amount of powder was welded onto the mill walls leading into loss of overall efficiency. The addition of stearic acid prevented the particles from welding to the vessel walls and helped to reduce the final particle size (Figure 4D,E). It’s worth to mention, that stearic acid is known to reduce the surface energy of the powder particles, preventing the formation of agglomerates during milling [38]. As a result, mutual collision between individual particles is possible, leading to the reduction in the size of processed particles.

The phase composition of prepared powders obtained by XRD is documented in Table 3 and Figure 5. In addition to the solid solution of Mg in Zn, intermetallic phases Mg_2_Zn_11_ and MgZn_2_ were also formed during mechanical alloying. The conditions for mechanical alloying were not achieved at combination of lower speed (400 RPM) and short time (10 min). Therefore, only pure Zn and Mg were observed in the powder mixture. Both pure powders were only accumulated together by cold welding. At 800 RPM, the deformation energy of the milling process increased significantly (622 kJ), which promoted particle interaction and caused the formation of new intermetallic phases. The microstructure of Zn-1Mg alloy consisted mainly of Zn solid solution and Mg_2_Zn_11_ phase. Besides, 1 wt.% of MgZn_2_ was detected. After milling at 1200 RPM, the microstructure was dominated by the intermetallic phase Mg_2_Zn_11_ in zinc matrix. Although the phase composition was similar during milling at 1200 RPM for 10 and 60 min, higher milling time in combination with stearic acid addition led to an increase in the homogeneity of the prepared powder and a decrease of particle size.

Mechanical alloying of Mg-6Zn powder mixture resulted in the formation of 65 wt.% of Mg_2_Zn_11_ phase and solid solution of Mg in Zn. Although the composition of the powder mixture corresponds directly to Mg_2_Zn_11_, a certain amount of Mg was dissolved in the Zn matrix, leading to the formation of oversaturated solid solution. In the case of Zn-16Mg, only MgZn_2_ phase was observed after mechanical alloying. Due to the brittleness of formed intermetallic phases, samples Zn-6Mg and Zn-16Mg were characterized by significantly reduced average particle size up to 2 and even 0.2 µm, respectively, after milling process (Figure 4).

### 3.2. Compaction by SPS

The microstructures of sintered materials are shown in Figure 6. The powder particles were covered with a thin oxide layer after mechanical alloying. This layer also remained in the microstructure after SPS compaction and created an interface between individual powder particles.

The Zn-1Mg and Zn-6Mg materials were characterized by minimal porosity (<0.3 vol.%) and high homogeneity of intermetallic phases. This is attributed to the presence of a soft zinc matrix, which is easily deformed during the SPS process, allowing the obtaining of materials with almost theoretical density. The Zn-16Mg alloys contained a significant portion of pores (21.4 vol.%). This material was entirely composed of the MgZn_2_ intermetallic phase, which is characterized by poor plasticity. During compaction, the particles are prone to shape changes, which leads to the increased porosity.

The Zn-1Mg and Zn-6Mg were characterized by a fine-grained microstructure. The average grain size of Zn-1Mg was 629 ± 274 nm, while 53 ± 25 nm was obtained for Zn-6Mg (Figure 7 and Figure 8). Magnesium is predominantly concentrated in the Mg_2_Zn_11_ phases (Figure 9), which are homogenously distributed in the zinc matrix and their average size is approximately 618 ± 269 nm and 58 ± 22 nm, respectively. The EDS measurements also confirmed the presence of Mg (≈0.4 ± 0.1 wt.%) in the zinc solid solution. It is worth mentioning that this estimation may be affected by the resolution of EDS measurements. Although the measurement of the concentration of Mg in the solid solution by EDS has to be taken carefully due to the extremely fine-grained microstructure and EDS resolution, the XRD result confirmed the obtained value. Precise Rietveld analyses of XRD data revealed the concentration of the Mg_2_Zn_11_ phase in the microstructure. By simple calculation, the 6 wt.% of Mg_2_Zn_11_ in the Mg-1Zn alloy corresponds to the 0.36 wt.% of Mg in Zn-1Mg. Subsequently, 0.54 wt.% of Mg remains to be incorporated in the zinc solid solution and oxide shells. Therefore, our results prove that during mechanical alloying, a non-equilibrium amount of Mg has been dissolved in the zinc solid solution and remained after processing by the rapid compaction using SPS.

It has been mentioned that the Zn-6Mg powder contained a portion of zinc solid solution, although the suggested composition directly relates to the Mg_2_Zn_11_ phase. To confirm this result, a detail of the microstructure (TEM) of the compacted Zn-6Mg alloy is shown in Figure 8. It is seen that the microstructure consists of extremely fine grains as well as intermetallic particles with the size about 53 ± 25 nm. It is worth to mention that similar grain sizes for Zn-based materials have been observed only for materials prepared by the high-pressure torsion (HPT) enabling significant decrease of the grain size but only for small affected area [39]. Figure 8a also shows residues of the oxide shells originating from the surface of powder particles with thickness between 50 and 80 nm.

### 3.3. Mechanical Properties

The hardness values measured according to the Vickers method are summarized in Table 4. The presence of intermetallic phases led to a significant increase in hardness values up to 271 HV1 determined for compacted Zn-6Mg. This material contained 62 vol.% of Mg_2_Zn_11_, which is characterized by a hardness value of 330 HV0.01 [39]. The slightly lower hardness value is related to the presence of a soft zinc matrix. Mg-16Zn is characterized by a lower hardness value compared to Zn-6Mg, but this is related to the extremely high porosity of the compacted material. The relatively high hardness values are also partly related to the extremely fine grain size. The values of the mechanical properties evaluated from the compression and bending tests are also summarized in Table 4. Examples of the compressive stress-strain curves are shown in Figure 10. Yield strength and/or ultimate strength obtained from the compression tests and the three-point bending tests differ significantly due to the different type of load applied to the test specimen. At the same time, this is related to the material defects (porosity, cavities, oxidic inclusions and shells, etc.) which more significantly affect the material response for the bending test. The ultimate strength of banded samples prepared by SPS ranged from 60 to 130 MPa, with the highest values being achieved for pure zinc prepared by SPS. It is evident that the presence of intermetallic phases in Zn-1Mg and Zn-6Mg as well as higher concentration of defects in these materials significantly reduced the flexural strength and plasticity of the materials. For this reason, the flexural yield strength was observed only for pure Zn. The other materials behaved almost completely brittle.

The microstructures of fracture surface after 3-point bending test are shown on the Figure 10. It can be clearly seen that fracture passes through brittle oxide shells or at the interface of oxide shall and matrix as documented by casual visibility of individual grains of material. Consequently, individual particles of original powder are visible at the fracture area.

Based on the compression behavior, compacted Zn and Zn-1Mg showed some plasticity (Figure 11). The compressive yield strength (CYS) and ultimate compressive strength (UCS) was increased with higher Mg content in the material, although the Zn-16Mg failed at 241 MPa due to high concentration of brittle intermetallic phases and incompletely sintered sample with high porosity.

## 4. Discussion

### 4.1. Microstructure

According to the Zn-Mg phase diagram, the solubility of magnesium in zinc at ambient temperature (25 °C) is low (0.008 wt.% [40]). Therefore, we selected mechanical alloying as the preparation technique. This method is known to dissolve elements in solid solution even at higher non-equilibrium concentrations. This has been observed also in our case, where the Mg concentration in the zinc solid solution was increased up to 0.4 ± 0.1 wt.%. Further, the selected combination of mechanical alloying with the right parameters (RPM, controlled temperature below 50 °C, stearic acid addition) and fast compaction by SPS resulted in an extremely fine microstructure with small intermetallic phases (Mg_2_Zn_11_, MgZn_2_).

Based on our experiments, a rotation speed of 1200 RPM (deformation energy 1012 kJ) was selected for the preparation of the powder, which was subsequently compacted by SPS. It is worth mentioning that mechanical alloying was also found to be successful even at 800 RPM (deformation energy 622 kJ), but resulted in a different phase composition. At 1200 RPM, increasing the MA time led to the generation of the desired phase composition. Furthermore, the selection of a higher rotational speed (Table 3) saved milling time and energy consumption.

Spark plasma sintering proved to be a suitable compaction method because the prepared materials preserved the extremely fine and homogeneous microstructure due to the rapid and intense compaction. The grain size of the prepared Zn-1Mg and Zn-6Mg materials reached 629 ± 274 nm and 53 ± 25 nm, respectively. This is almost impossible to achieve for zinc characterized by a very low recrystallization temperature (≈−12 °C [41]) by conventional techniques. Pure as-cast zinc usually possesses a grain size in hundreds of µm [9,13]. The addition of magnesium caused the reduction in the grain size to about 150 µm for 1 wt.% of Mg [9,14] and to 48 µm for 3 wt.% of Mg [9]. In addition, thermomechanical processing (extrusion, rolling, drawing, etc.) is generally performed to further decrease the grain size of the as-cast materials. For example, a Zn-1Mg alloy was extruded from as-casted ingot with an extrusion ratio of 16:1 and a temperature of 300 °C to obtain material with an average grain size equal to 5 µm [17]. The Equal Channel Angular Pressing (ECAP) is a well-known technique of intensive plastic deformation that allows the extreme refinement of the microstructure due to the dynamic recrystallization processes taking place during repeated passes through the die. By the selection of this technique, the average grain size of 2 μm was observed for the Zn-3Mg alloy prepared from as-casted ingot [42]. The highest grain refinement for Zn-Mg alloys with the only comparable grain size value (590 ± 60 nm) was achieved by high-pressure torsion (HPT) [39]. However, this preparation method has a significant limitation in the size of the affected area.

The Mg_2_Zn_11_ and MgZn_2_ intermetallics achieved a similar size to the matrix grains (≈ 612 ± 269 nm, 58 ± 22 for Zn-6Mg). That is much finer than in the case of as-casted or even thermomechanically processed Zn-1Mg alloys [43,44]. Gong et al. [44] observed the intermetallic phases in the as-cast alloy with the size of tens of µm. By extrusion, the microstructure became much finer and broken intermetallic phases reached the size of several µm causing the improvement of mechanical properties. Kubásek et al. successfully broke the large Mg_2_Zn_11_ phases of the as-cast Zn-0.8Mg-0.2Sr alloy by extrusion, although the average particle size was still about 1.5 μm [43].

Several publications have dealt with Zn-based alloys prepared by powder metallurgy. Ali et al. [28] and Guleryuz et al. [45] used mechanical alloying in planetary ball mills with much less efficient parameters (350 RPM, 4 h [28]; 250 RPM, 8 h [45]) and compaction by traditional sintering (350 °C, 4 h, 300 MPa [28]; 410 °C, 30 min, 30 MPa [45]) to prepare a Zn-Mg alloy of similar or close composition. Prepared Zn-1Mg was characterized by a homogeneous microstructure and similar phase composition but also an extremely coarse microstructure including a large grain size (10–100 µm) and coarse intermetallic phases (≈ 50 µm). Increasing the amount of Mg (3 and 6 wt.%) caused the segregation of the Mg_2_Zn_11_ phase and generation of an inhomogeneous microstructure.

### 4.2. Mechanical Properties

The obtained mechanical properties are affected by several factors. The high strengthening effect was achieved by precipitation of the Mg_2_Zn_11_ (MgZn_2_) intermetallic phases, which were homogeneously distributed as fine particles in the Zn-1Mg and Zn-6Mg materials. This led to a significant increase in hardness from 38 HV1 for pure zinc to 123 HV1 for Zn-1Mg and 271 HV1 for Zn-6Mg. As a result of the high porosity and insufficient conditions for complete sintering, there was no further increase in the hardness value for the Zn-16Mg alloy, which was composed of a nearly pure MgZn_2_ intermetallic phase with an estimated hardness of 445 HV [39]. The effect of this contribution is obvious also from CYS values reaching 86 and 361 MPa for Zn and Zn-1Mg. For the estimated values of the strengthening contribution of the intermetallic phases, Equations (3)–(5) may be adopted from the literature [15,46,47,48]. In these relations, *σ_p_* represents the contribution of the intermetallic phases (Mg_2_Zn_11_) to the CYS values, ∅ is the modulus correction factor, *γ* is the accommodation factor, *µ^M^* is the shear modulus of the zinc matrix (50 GPa), *µ^P^* is the shear modulus of the Mg_2_Zn_11_ phase (33 GPa), V represents the volume fraction of the Mg_2_Zn_11_ phase in the material (7 vol.% for Zn-1Mg, 65 vol.% for Zn-6Mg), *ε* corresponds to the unrelaxed plastic strain (estimated as 0.007 according to the literature [15]) and *ν* is Poisson ration of Mg_2_Zn_11_ (0.29).
(3)$$\sigma_{p}=4\;\emptyset \;\gamma \;\mu^{M}\;V\;\varepsilon$$
(4)$$\gamma =\frac{1}{2\left(1-\nu \right)}$$
(5)$$\emptyset =\frac{\mu^{P}}{\mu^{P}-\gamma \left(\mu^{P}-\mu^{M}\right)}$$

Based on this calculation, the effect of Mg_2_Zn_11_ on the CYS value corresponds to 50 MPa and 470 MPa for Zn-1Mg and Zn-6Mg, respectively.

Due to the presence of Mg partially dissolved in Zn (≈0.4 wt.%), the solid solution strengthening mechanism contributes to the mechanical properties. However, this strengthening effect is considered to be relatively low. Liu et al. [49] studied the strengthening mechanisms of Mg in the as-cast Zn-Mg alloys. They pointed out that the contribution of solid solution strengthening for 0.03 wt.% of Mg dissolved in the Zn matrix was around 12 MPa. In this study, the measured concentration of Mg in solid solution corresponds to the 0.4 wt.%, which is 13× higher, but according to the calculation performed by Chen et al. [50] the dependency on Mg concentration is very weak and the estimated value is ≈15 MPa.

Therefore, in the case of Zn-1Mg, 50 MPa is considered for the strengthening effect of Mg_2_Zn_11_, 15 MPa for solid solution strengthening and 296 MPa remains for other strengthening contributions. Based on the XRD measurements, negligible residual stress was observed in the Zn-1Mg processed by SPS. Therefore, the strengthening by grain size is considered as the main factor affecting the obtained mechanical properties. However, it is worth mentioning that dispersed oxide particles were observed in the material. These particles are extremely fine (100 nm) and difficult to quantitatively determine. However, they are presented dominantly at the grain boundaries, which rather helps to stabilize the fine-grained microstructure (suppressed coarsening of grains) and prevents the slip of grain boundaries [51,52] than strengthening the material by direct interaction with dislocations. Therefore, we expect that the suggested contribution of 296 MPa is related especially to the grain size strengthening effect.

Unfortunately, the biggest issue of the prepared materials was the particle interface in the compacted products, which was occupied by thin oxide shells (50–80 nm). This phenomenon was also observed in the work of Krystýnová et al. [27]. The presence of oxides leads to an increase in the stress in their surroundings and a significant reduction in the materials’ ductility. Crack propagation is much easier in the brittle oxide phase than through the deformable zinc matrix.

We assumed that the oxides on the surface of the powder particles form a network-like structure after compaction. However, the grinding process was carried out in an inert argon atmosphere. Between MA and SPS, the powder was exposed to the atmosphere and the particles could oxidize. Another consideration is the condition of the original powder, which also had surface oxidation.

The effect of oxide shells has also been observed in other compaction methods [18,53]. Samples containing an oxide network showed the intergranular fracture mechanism (propagation of the crack around particles through oxide layer), which decreases overall ductility. A similar effect of presented oxides shells in the microstructure after SPS was observed for the Mg [54] and its alloys [55,56]. It has been discussed that SPS could self-clean the surface of the input powder [57], but we did not observe such behavior. The existence of a brittle oxide network can be disrupted by following thermomechanical processing such as extrusion or rolling. Such processing is suggested to further improve the mechanical properties by formation of a composite-like material.

Several studies dealt with the synthesis, processing and characterization of Zn-Mg materials by powder metallurgy [28,58,59]. The obtained results indicated that these procedures led to improved mechanical properties due to the fine microstructures, precipitations of intermetallic phases and solid solution strengthening, although the presented results are still far from the values obtained in presented paper (Table 5). Yang et al. [58] prepared Zn-1Mg powder from pure metals in a planetary ball mill (260 RPM, 4 h) with subsequent compaction by Selective Laser Melting (SLM). They observed partially dissolved magnesium in the Zn matrix according to the EDS analyzer. At the same time, significant precipitation strengthening was observed, caused by uniformly precipitated eutectic mixture along grain boundaries.

Yan et al. [59] reported the grain refinement of the Zn-1Mg alloy to 7.3 µm, however, their mechanical alloying technique (mixing for 8 h in three-dimensional blender) was not efficient enough because the residual magnesium phase remained in the material. Ali et al. [28] used mechanical alloying in planetary ball mills (350 RPM, 4 h) to prepare Zn-1Mg with a homogeneous microstructure and observed strengthening due to the Mg_2_Zn_11_ or MgZn_2_ phases (increase in compressive strength from 100 MPa of pure Zn to 178 MPa of the Zn-1Mg alloy). No traces of solid solution were observed in the matrix, but the authors observed particles of residual pure Mg. The addition of 3 wt.% of Mg or even more to Zn led to the segregation and the formation of inhomogeneous microstructures, which caused significant deterioration of mechanical properties (Table 5). Additionally, high porosity remained in the material after compaction (cold pressing 350 MPa and sintering at 350 °C for 4 h) contributing to the low strength.

Compared to the several mentioned studies, the Zn-1Mg prepared in this work is characterized by much finer and more homogeneous microstructures with uniformly distributed fine intermetallic phases. At the same time, the Zn-1Mg was denser due to the selected compaction method—SPS. All these factors resulted in superior values of hardness and UCS (Table 5). It is worth mentioning that the UCS of Zn-1Mg is about 80 MPa higher than that for the same material prepared by conventional casting (285 MPa [19]), hot extrusion (≈310 MPa [15]) or even hydrostatic extrusion (≈390 MPa [15]). In the case of the three-point bending tests, the effect of brittle oxide shells in connection with the presence of the high-volume fraction of intermetallic phases and some residual porosity within the material has serious consequences in low values of ultimate flexural strength.

Zn-6Mg and Zn-16Mg were sufficiently prepared in the form of powders by mechanical alloying with the final phase composition reflecting the dominance of the Mg_2_Zn_11_ and MgZn_2_ intermetallic phases in microstructures, respectively. However, at the same time, it has been shown that due to the high content of the brittle intermetallic phase, the compaction by SPS is complicated with high residual porosity in the material, leading to the poor mechanical properties of prepared products. However, it is worth mentioning that the possibility to prepare extremely fine particles of Mg_2_Zn_11_ and MgZn_2_ by mechanical alloying may be used in the development of composite materials by mixing with other metallic powders.

Based on the results, Zn-1Mg could be used for a variety of low load-bearing orthopedic applications or biomedical devices that are mainly stressed at compression due to the similarities with bioceramics that are used such as bone plates or screws [36].

## 5. Conclusions

The Zn-Mg binary alloys containing 1, 6 and 16 wt.% of Mg were prepared by a combination of powder metallurgy techniques including mechanical alloying and fast compaction technique–spark plasma sintering. The combination of these methods led to the preparation of the nanograined Zn-1Mg alloy containing zinc grains with an average grain size 618 ± 269 nm and the intermetallic phase Mg_2_Zn_11_ of the similar size. Additionally, a nonequilibrium amount of Mg (≈0.4 wt.%) was dissolved in the zinc solid solution. These factors led to the high values of CYS (361 MPa) and UCS (436 MPa) of Zn-1Mg alloys. After the addition of 6 wt.% of Mg, the Mg_2_Zn_11_ phase dominated the microstructure leading to the increase in UCS (574 MPa) but the disappearance of plasticity. Zn-16Mg (wt.%) was completely consisted of the MgZn_2_ phase, leading to the high porosity after SPS as a consequence of the lack of plasticity of powder particles. This resulted in poor strength (245 MPa). Although the Zn-1Mg compacted product has been characterized by poor plasticity, especially due to the presence of thin oxide shells on original powder particles, extremely fine-grained and homogenous microstructures and high values of strength predetermine this material for applications like biodegradable medical devices, which are preferably loaded under compression.

## Figures and Tables

**Figure 1 materials-15-05272-f001:**
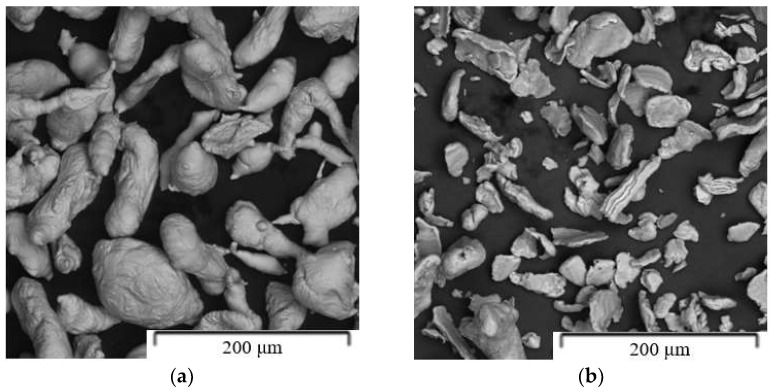
Shape and morphology of the initial powders: (**a**) Zn, (**b**) Mg.

**Figure 2 materials-15-05272-f002:**
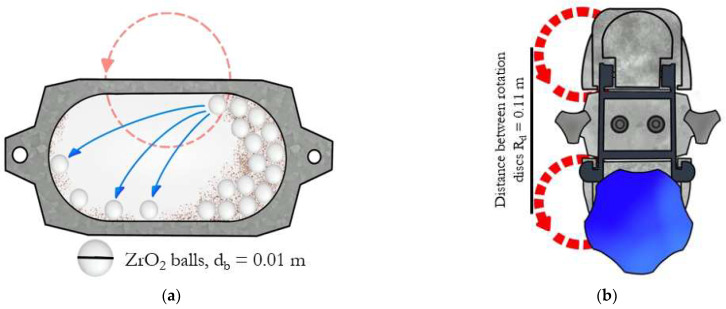
Retch E-max mill: (**a**) Vessel, (**b**) Clamping system.

**Figure 3 materials-15-05272-f003:**
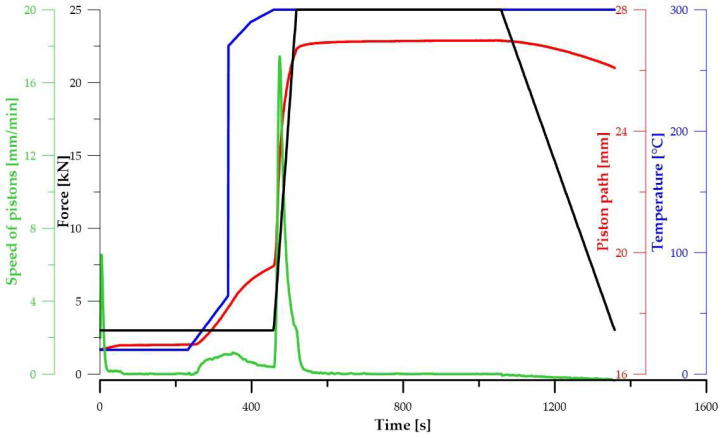
Time profiles of the SPS process conditions.

**Figure 4 materials-15-05272-f004:**
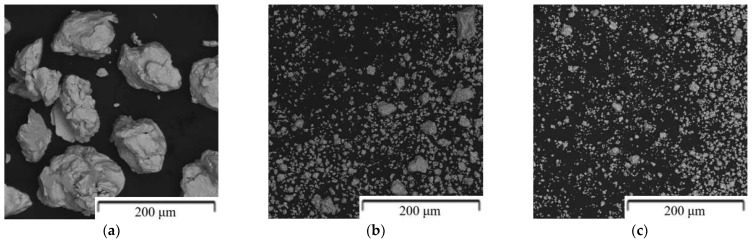
Shape and morphology of the powders prepared by mechanical alloying: (**a**) Zn-1Mg (1200 RPM, 60 min), (**b**) Zn-6Mg (1200 RPM, 60 min), (**c**) Zn-16Mg (1200 RPM, 60 min), (**d**) Zn-1Mg (1200 RPM, 30 min), (**e**) Zn-1Mg-0.08g stearic acid (1200 RPM, 30 min).

**Figure 5 materials-15-05272-f005:**
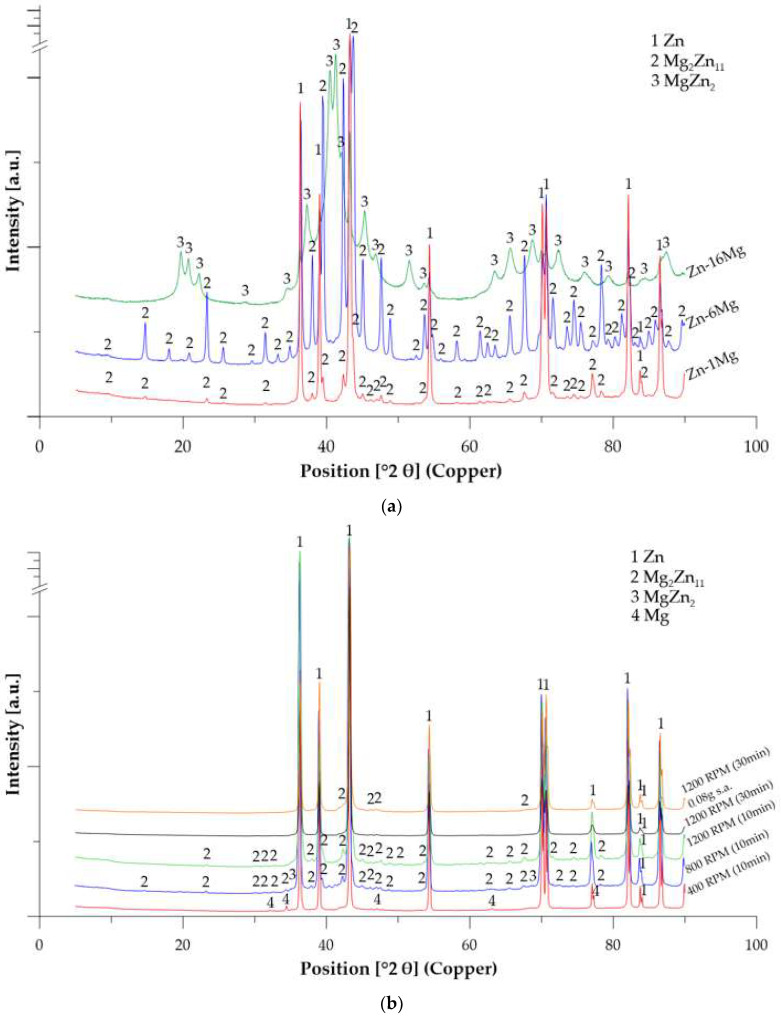
XRD diffractograms of mechanically alloyed materials: (**a**) Zn-xMg (x = 1, 6, 16 wt.%) powders milled at 1200 RPM for 1 h, (**b**) Zn-1Mg powders milled under different conditions.

**Figure 6 materials-15-05272-f006:**
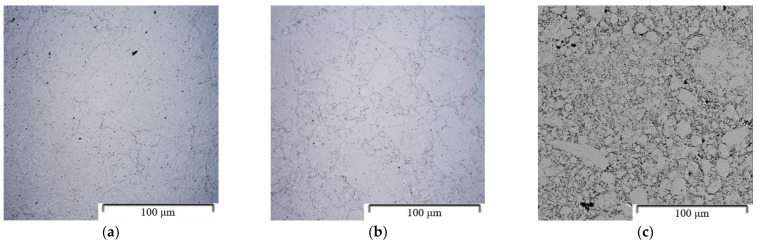
Materials compacted by SPS (OM): (**a**) Zn-1Mg (1200 RPM, 60 min), (**b**) Zn-6Mg (1200 RPM, 60 min), (**c**) Zn-16Mg (1200 RPM, 60 min).

**Figure 7 materials-15-05272-f007:**
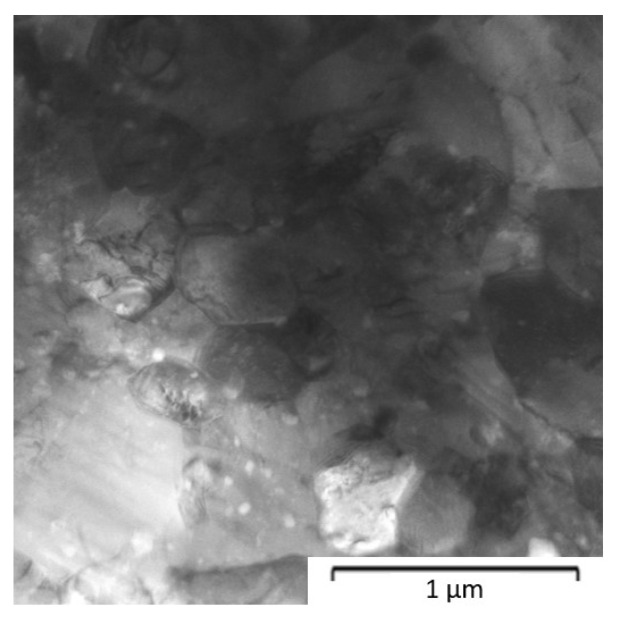
Microstructure of Zn-1Mg alloy compacted from mechanically milled powder (1200 RPM, 60 min) by SPS–TEM.

**Figure 8 materials-15-05272-f008:**
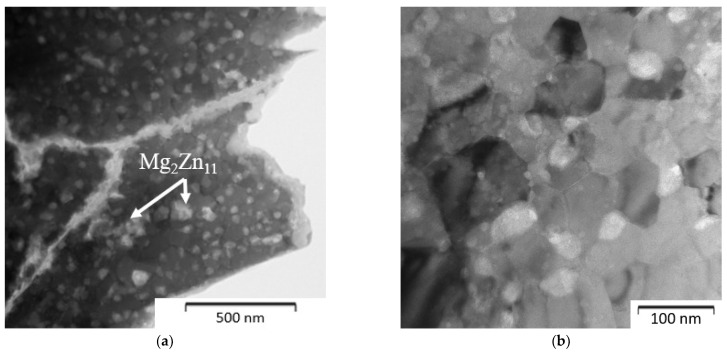
Microstructure of Zn-6Mg alloy compacted from mechanically milled powder (1200 RPM, 60 min) by SPS–TEM: (**a**) view of the powder particle, (**b**) detailed view of the internal structure.

**Figure 9 materials-15-05272-f009:**
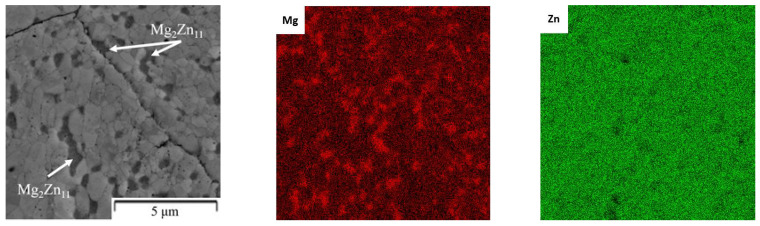
The distribution of elements in the Zn-1Mg alloy compacted by SPS (powder was prepared at conditions 1200 RPM, 60 min)–EDS (SEM).

**Figure 10 materials-15-05272-f010:**
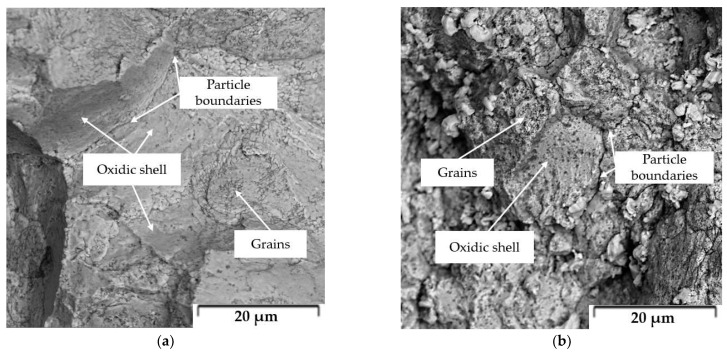
Fractography of samples after 3-point bend test: (**a**) Zn-1Mg, (**b**) Zn-6Mg.

**Figure 11 materials-15-05272-f011:**
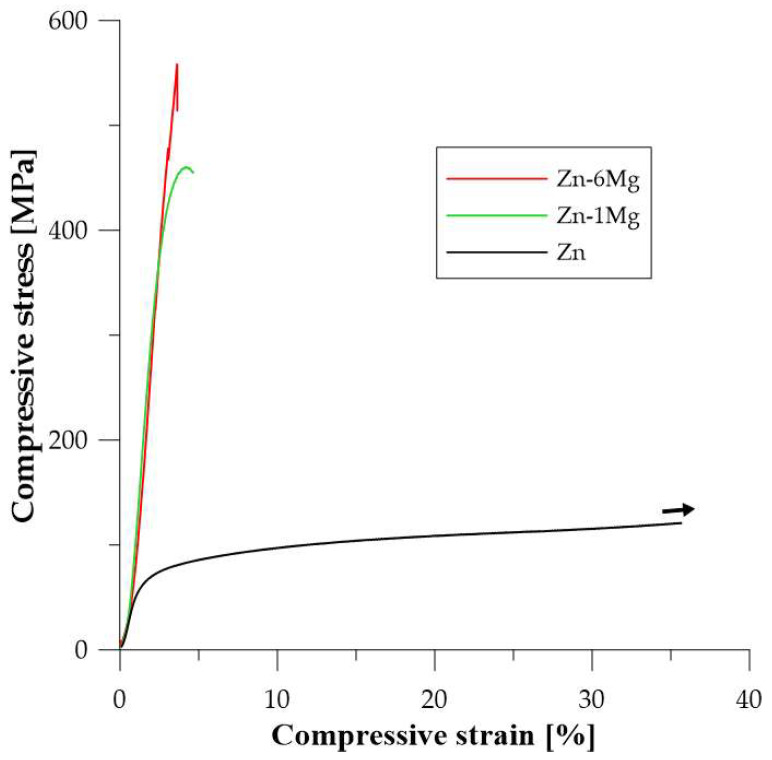
Compressive stress-strain curves of Zn-based materials prepared by mechanical alloying and SPS. Black arrow represents the continuation of sample deformation without fracture.

**Table 1 materials-15-05272-t001:** Summary of mechanical alloying conditions.

Composition of the Alloy	Amount of Stearic Acid [g]	RPM [Rotation/min]	Time [min]	Balls/Powder Weight Ratio	Selected for Compaction
Zn	-	-	-	-	✔
Zn-1Mg	-	400	10	10:1	-
Zn-1Mg	-	800	10	10:1	-
Zn-1Mg	-	1200	10	10:1	-
Zn-1Mg	-	1200	30	10:1	-
Zn-1Mg	0.08	1200	30	10:1	-
Zn-1Mg	0.08	1200	60	5:1	✔
Zn-1Mg	0.08	800	120	5:1	-
Zn-6Mg	0.08	1200	60	5:1	✔
Zn-16Mg	0.08	1200	60	5:1	✔

**Table 2 materials-15-05272-t002:** Calculated angular speed and deformation energy based on milling parameters.

RPM [Rot./min]	Angular Speed *ω_d_* [Rad/s]	Deformation Energy *E_C_* [kJ]
400	42	271
800	84	622
1200	126	1012

**Table 3 materials-15-05272-t003:** Phase composition of the Zn-Mg alloys prepared by mechanical alloying (XRD analyses).

Composition (Weight of Milling Balls: Powder)	RPM [Rot./min]	Milling Time [min]	Zn [wt.%]	Mg_2_Zn_11_ [wt.%]	MgZn_2_ [wt.%]	Mg [wt.%]
Zn	-	-	100	-	-	-
Mg	-	-	-	-	-	100
Zn-1Mg (10:1)	400	10	99			1
Zn-1Mg (10:1)	800	10	94	5	1	-
Zn-1Mg (10:1)	1200	10	94	6		-
Zn-1Mg (5:1)	1200	30	100			-
Zn-1Mg (5:1) *	1200	30	99	2		<1
Zn-1Mg (5:1) *	800	60	97	3	-	<1
Zn-1Mg (5:1) *	1200	60	94	6		-
Zn-1Mg (5:1) *	800	120	93	7		-
Zn-6Mg (5:1) *	1200	60	33	65	2	-
Zn-16Mg (5:1) *	1200	60	6	-	94	-

* The addition of 0.08 g Stearic acid.

**Table 4 materials-15-05272-t004:** Mechanical properties of materials compacted by the SPS.

Composition	RPM/Milling Time [min]	HV1	Compression Test		3-Point Bend Test		
			σ_CYS_ [MPa]	σ_UCS_ [MPa]	σ_YS_ [MPa]	σ_US_ [MPa]	Strain [%]
Zn	-	38 ± 2	86 ± 2	-	104 ± 13	136 ± 12	0.73 ± 0.11
Zn-1 Mg *	1200/60 min	123 ± 2	361 ± 3	436 ± 6	-	59 ± 5	0.07 ± 0.01
Zn-6 Mg *	1200/60 min	271 ± 12	-	574 ± 29	-	74 ± 15	0.02 ± 0.01
Zn-16 Mg *	1200/60 min	222 ± 12	-	241	-	-	-

* The addition of 0.08 g Stearic acid.

**Table 5 materials-15-05272-t005:** Mechanical properties of zinc and its alloys prepared by powder metallurgy.

Composition	Preparation	Hardness	σ_CYS_ [MPa]	σ_UCS_ [MPa]	E [%]	Ref.
Zn	MA & SPS	38 ± 2	86 ± 2	-	-	This work
Zn-1Mg		123 ± 2	361 ± 3	436 ± 6	-	
Zn-6Mg		271 ± 12	-	574 ± 29	-	
Zn-1Mg	MA, Cold press, sintered, forged	81 ± 5	-	245 ± 12	5.6 ± 1.4	[58]
Zn	MA, Hydrostatic pressing & sintering	40 ± 1	-	100 ± 35	-	[28]
Zn-1Mg		60 ± 23	-	178 ± 13	-	
Zn-6Mg		41 ± 1	-	122 ± 10	-	

σCYS = compressive yield strength; σUCS = ultimate compressive strength; and E = elongation.

## Data Availability

The data are available at the email addresses of the corresponding authors.

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
