# Peer review of "Advanced Zinc–Magnesium Alloys Prepared by Mechanical Alloying and Spark Plasma Sintering"

_materials, 2022, doi:10.3390/ma15155272_

Round 1

Reviewer 1 Report

Dear Authors additional comments are obligatory to your work.

  1. Sigma means density, surface density and contributions of the intermetallic phases, therefore Authors have to use additional different symbols for considered meanings. (page 4 and page 13)
  2. One units for temperature should be use in the paper.
  3. Authors should explain, does 10 min of SPS was started after obtained required level of temperature - 573 K or this time include period of heating.
  4. Sketch of SPS system should be add to work.
  5. Height of sample before and after SPS should be add to paper. Moreover data of sample weight is important for Readers.
  6. Sketch from figure 2b is not needed, maybe mill photograph will be better. Name of mill manufacturer should be improve.
  7. Comments on MA and SPS trial repetitions should be include, especially for data on results errors.
  8. What means SPS-OM? (figure 5)
  9. Mechanism of thin oxide layer formation after MA should be explain.
  10. First line, page 9 – mistake in the sentence.
  11. XRD, EDS, TEM analysis are perform for specific area of sample. Please explain your location of area of analysis in the samples.
  12. What means laboratory temperature? (page 11)
  13. Expression 3 from page 13 is not explain in the text of paper.

Reviewer 2 Report

Some comments should be considered by the authors before accepted. 

1.  Renumbering of equations. page 4, eqs (1-2) and page 13, eqs (1-3)

2. How to convent from wt.% to vol.% of Mg2Zn11 and MgZn2?

3. Page 4, σ is ρ

4. Page 4. The authors selected the powder mixtures based on the smallest particle size distribution and the most suitable phase composition. Please give the particle size distribution in the table. What is the most suitable phase composition?

5. No description of figure 2 and figure 7 in the main text.

6. What is the meaning of average particle size? In the main text, the unit is μm, while in the table 3, the unit is μm2.

7. What is the milling time for the Zn-16Mg sample? Two different values, 60 and 120 min are shown in the manuscript.

8. Are the black dots in Figure 5 are pores? If yes, the pores in figure 5a is more than those in figure 5b.

9. Page 9. (Figure 7 andError! Reference source not found.) please correct it.

10. What are the meanings of CYS and UCS?

11. What are the difference of two “Yield Strength” in the Table 4?

12. Page 12. Zn-0,8Mg-0,2Sr and Mg (3, 6 wt. %). Use dot instead of comma.

13. Page 13. How to obtain the volume fraction of Mg2Zn11 phase in the material? Why did the authors use one value for two materials (Zn-1Mg and Zn-6Mg)?

14. What is the meaning of hm. %?

15. What are the meanings of σCYS, σUCS and E in the table 5?

16. In conclusions, “intermetallic phase Mg2Zn11 of the same size”. In the manuscript, it is not the same size.

Reviewer 3 Report

This work examines biological materials produced from zinc-magnesium alloys using the MA and SPS techniques. Since then, the authors have discovered that Zn-1Mg alloy possesses good mechanical properties, biocompatibility, and biodegradability.

In the introduction, the authors evaluated the existing biomedical materials, focusing on Zinc and its alloys. The authors explain that Zinc has a good corrosion rate and does not produce harmful corrosion products. To increase the mechanical properties of pure Zinc, it is required to locate an alloying element and a composition containing that element. The authors have demonstrated, through extensive research and analysis, that Mg is a good element for enhancing the desirable qualities of biomedical materials in this study. The powder metallurgy process yields a rather homogeneous product that is subsequently subjected to SPS for completion. Finally in this section, the objective of the paper is to learn about nanoparticles and the optimal milling mode. This introduction is provided sequentially with closely related sections, improving comprehension of the study's objective.

The experimental section describes the powder mixture's composition, powder weight, PCA, and milling modes (Table 1). Table 2 presents the deformation energy calculation based on the collision energy of the grinding ball and the angular speed. The milled powder sample is then analyzed for XRD, microstructure (SEM, EDS, TEM). Anh the bulk sample by SPS is examined for mechanical properties. In this section, the authors have outlined the relevant process parameters in detail. However, there are some questions: is the 1 kgf load in the hardness test reasonable? Add the amount of stearic acid proportional to how much percent of the powder in the grinding via? What is the grinding Via's volume? since this is an additional cause for concern.

The experimental section discusses the composition, powder weight, PCA, and milling modes of the powder mixture (Table 1). The deformation energy calculation is presented in Table 2 based on the collision energy of the grinding ball and the angular speed. The sample of milled powder is next tested for XRD and microstructure (SEM, EDS, TEM). The bulk samples by SPS were evaluated for its mechanical properties. In this section, the authors have outlined the relevant process parameters in detail.. However, certain questions remain: Is the 1 kgf load appropriate for the hardness test for the low HV values?. What is the grinding Vessel's volume? since this is a further cause for concern.

In the third section, the results of the grinding modes are presented and contrasted, revealing varied grain morphologies with various alloy compositions and also with the addition of stearic acid. The problem may lay in table 3, as there are so many parameters. How do you determine percent Zn and phase fraction for each grinding mode and XRD pattern, using the phases listed in table 3?. The SPS technique then produces the intermetallic phases depicted in Figure 5. And how do the authors calculate the particle size of 629 +/- 274 nm using ImageJ (evaluation techniques)? This should be addressed and clarified to provide the reader with further information regarding the post-sintering sample. References for the assignment of XRD diffractions to the elements and intermetallics in discussion in Figure 4 should be provided.

The discussion section provides the author's reasons and arguments, and via reference to past studies, likely, using a Zn-1Mg alloy, a material with small grain size, homogeneous microstructure, and uniform intermetallic phases can be produced. On the page 8, How do you confirm this statement “The Zn-1Mg and Zn-6Mg were characterized by a homogeneous fine-grained microstructure”?. In addition, the highest mechanical properties are achieved with the combination of the MA and SPS processes: Good Hardness (123 HV1) and the highest critical yield strength of 361 MPa. Although Zn-6Mg is twice as hard as Zn-1Mg, it is not very homogeneous (high porosity).

The discussion section provides the author's justifications and arguments, and by referencing prior research, it is likely that a material with fine grain size, homogenous microstructure, and uniform intermetallic phases may be manufactured using a Zn-1Mg alloy. How can you verify the statement on page 8 that "The Zn-1Mg and Zn-6Mg were characterized by a homogeneous fine-grained microstructure"? In addition, the combination of the MA and SPS processes achieves the highest mechanical properties: Good hardness (123 HV1) and the highest yield strength (361 MPa). Although Zn-6Mg is twice as hard as Zn-1Mg, it is not very homogeneous (high porosity).

Reviewer 4 Report

In the research paper submitted by the author, the alloying of zinc and magnesium powders and the preparation of zinc magnesium alloy ingots by spark plasma sintering were studied. The microstructure and mechanical properties of ingots were analyzed. This study has some novelty, but it still has room for improvement. Here are some suggestions for the author's reference.

1. When discussing the results of powder preparation, the role of stearic acid was mentioned. Whether stearic acid was used in this experiment and its dosage were suggested that the author should give corresponding data.

2. It can be found from table 3 that there is a large loss in the content of magnesium in the preparation of alloy powder. It is suggested that the author explain the reason.

3. The representation of Figure 4 may be inappropriate. The author shows different curves in the same coordinate system, and vertically staggers the curves, but the ordinate should belong to only one curve. It is suggested that the author check and modify the coordinates.

4. It is mentioned in the article that an oxide layer is formed on the surface of the powder, and this oxide layer is finally retained between the grains, so why not add protective gas to prevent the powder oxidation during the processing? In addition, the oxygen content in powder and ingot and the form of oxide are not analyzed in this paper. It is suggested to supplement and prove the destruction of oxide network mentioned by the author in the discussion.

5. The quotation in subsection 4.1 of the discussion part may be inappropriate. The quotation in this part should quote phenomena similar to those in this article to analyze the causes of the phenomenon, and the author's quotation method is more suitable to appear in the introduction. It is suggested to make some adjustments.

6. It is suggested that the author add the characterization of the fracture of the failed parts to analyze the failure form of the ingot. In this way, the oxide layer network inside the ingot can also be further analyzed.

Round 2

Reviewer 3 Report

The revised manuscript is improved significantly.

The particle size distribution values in table 3 is not reflected powder's characteristics. This particle size distribution should be measured using laser particle size analyzer. I think this column should be removed because the images in Fig. 1 and Fig. 4 show particle size characteristics.

Author Response

Dear Reviewer,

Thank you for your comment. Edits have been made to the manuscript.